# Sustainable Cellulose-Aluminum-Plastic Composites from Beverage Cartons Scraps and Recycled Polyethylene

**DOI:** 10.3390/polym14040807

**Published:** 2022-02-19

**Authors:** Irene Bonadies, Roberta Capuano, Roberto Avolio, Rachele Castaldo, Mariacristina Cocca, Gennaro Gentile, Maria Emanuela Errico

**Affiliations:** 1National Research Council of Italy, Institute for Polymers Composites and Biomaterials (IPCB-CNR), Via Campi Flegrei 34, 80078 Pozzuoli, Italy; irene.bonadies@cnr.it (I.B.); roberta.capuano@ipcb.cnr.it (R.C.); rachele.castaldo@ipcb.cnr.it (R.C.); cocca@ipcb.cnr.it (M.C.); gennaro.gentile@cnr.it (G.G.); mariaemanuela.errico@cnr.it (M.E.E.); 2Department of Mechanical and Industrial Engineering—DIMI, University of Brescia, Via Branze 38, 25121 Brescia, Italy

**Keywords:** recycling, beverage cartons, composites, polymer processing, cellulose

## Abstract

The sustainable management of multilayer paper/plastic waste is a technological challenge due to its composite nature. In this paper, a mechanical recycling approach for multilayer cartons (MC) is reported, illustrating the realization of thermoplastic composites based on recycled polyethylene and an amount of milled MC ranging from 20 to 90 wt%. The effect of composition of the composites on the morphology and on thermal, mechanical, and water absorption behavior was investigated and rationalized, demonstrating that above 80 wt% of MC, the fibrous nature of the filler dominates the overall properties of the materials. A maleated polyethylene was also used as a coupling agent and its effectiveness in improving mechanical parameters of composites up to 60 wt% of MC was highlighted.

## 1. Introduction

Plastic-paper multilayer materials find wide application in the packaging sector. Among the most common examples of such systems are the so-called beverage cartons or multilayer cartons (MC). They are widely used for the storage of dairy products, juices and many other liquid foods, such as pre-cooked vegetables and soups.

MCs are constituted by a structural paperboard core sandwiched between plastic and aluminum layers with barrier and sealing functions (Figure 1) [1]. Non aseptic MCs for dairies are constituted only by paper (79 wt%) and polyethylene (PE, 21 wt%) layers, while an aluminum foil (5 wt%, partially substituting paper) is used to provide high protection against light and oxygen is used in aseptic cartons for long shelf-life products (UHT milk, juices).

With the increasing awareness on the environmental and management issues caused by single use disposable materials (i.e., by definition, most packaging materials), a large effort has been devoted to increase the collection and recycling rate of packages, including MCs. The constituents of multilayer cartons (high quality paper, virgin PE, and aluminum foil) are fully recyclable materials. From a theoretical point of view, MCs can therefore be recycled. However, their composite nature implies a non-straightforward separation of the different layers in order to have an effective recycling. Paper mills are currently the usual destination of waste MCs, as paper accounts for 75–80 wt%, but the pulping conditions used to recycle regular wastepaper are not suited to efficiently separate the cellulose from PE/Al layers in MCs. In fact, dedicated pulping stations must be implemented for the effective recovery of cellulose fibers from MCs [2,3]. Consequently, a separated collection and dedicated processing lines for such materials is required, strongly limiting the recycling rates of MCs [2]. The byproducts-including polyethylene, aluminum, and a variable amount of residual cellulose-are usually dried, ground, and processed to obtain composites.

The problems related to the recycling of multilayer paper/plastic materials are expected to increase in the next few years due to the increased use of paper-based packaging materials [4]. This is in part driven by the ban on some single use plastic items recently enforced in the EU [5] which is boosting the use of alternative solutions, such as plastic-lined paper dishes and cups.

As an alternative to dedicated collection and pulping, the grinding of MCs to obtain a (mainly) cellulosic filler employed for the fabrication of polymer-based composites has been investigated by several authors (see review [3] and references therein). The realization of MC-based composites can represent a convenient recycling route as it does not require layers’ separation. The resulting materials will show reasonable properties and low cost, with foreseen applications similar to well-known wood-plastic composites [6,7]. Polyolefins are the most investigated matrices for such composites. However, different polymers have been proposed, including thermosets. Due to the highly polar and hydrophilic nature of cellulosic materials, the ability of the selected polymer to form strong, well adhered interfaces with such fillers is a key factor to define processing and additivation strategies [8,9]. Polar matrices such as polyvinyl alcohol showed good adhesion and improved mechanical response when reinforced with MC [10], while studies on HDPE composites filled with up to 60 wt% of milled MCs [6] pointed out the need for an intense mixing to help separate paper fibers and the important role of coupling agents in improving polymer/fibers adhesion.

In this paper, these concepts are further developed. MC based materials have been investigated over a wide compositional range, demonstrating the effective production of panels with up to 90 wt% MC and extending the investigation to cartons containing aluminum foil (Al). A recycled PE was employed as the polymeric phase, leading to the realization of fully recycled, sustainable composites. Thermal, mechanical, and water sorption properties were investigated as a function of composition, and the effectiveness of a maleated polyethylene coupling agent was pointed out.

## 2. Materials and Methods

### 2.1. Materials

Non aseptic (C) and aseptic (C-Al) pre-consumer carton scraps were kindly provided by Italpack Cartons S.R.L., Lacedonia, Italy. Post-consumer high density polyethylene (PE) flakes were kindly supplied by a local selection platform. Maleated linear low-density polyethylene (MAPE, density 0.92 g/cm^3^ grafted maleic anhydride 1 wt%) was kindly supplied by Agricola Imballaggi S.R.L., Pagani, Italy.

### 2.2. Composites Preparation

Both C and C-Al cartons were finely ground by means of a Retsch SM100 rotary knife mill (Retsch GmbH, Haan, Germany) with a 1 mm bottom sieve. The obtained powders were then mixed with PE at 175 °C in a Brabender Plastograph internal mixer (Brabender GmbH and co KG, Duisburg, Germany), equipped with a 55 cm^3^ mixing chamber and two counter-rotating blades. The following procedure was adopted. First, the appropriate amount of polymeric phase (PE + MAPE) was introduced in the chamber and allowed to melt at a reduced speed for 1 min. Then, the MC powder was slowly added. The chamber was then sealed and the speed was raised to 40 rpm, mixing the materials for a further 8 min.

After melt mixing, the resulting materials were allowed to cool at room temperature and pelletized using hand cutters. They were then compression molded by means of a Collin P200 hot press equipped with a water circulated cooling system (COLLIN Lab and Pilot Solutions GmbH, Maitenbeth, Germany), using a temperature of 180 °C, a pressure of 50 bar and a permanence time of 5 min, followed by cooling to room temperature that was obtained in approximately 5 min. Sheets with a thickness of either 1 or 3 mm were obtained.

### 2.3. Characterization

The particle size distribution of the milled MC powders was determined by sieving through a stack of metal sieves (FILTRA Vibracion, Badalona, Spain) with nominal mesh sizes of 1000, 500, 200, 100, and 50 µm, and using a Retsch AS 200 vibratory sieve shaker (Retsch GmbH, Haan, Germany).

Morphological analysis was carried out on impact-fractured surfaces by means of a Phenom compact scanning electron microscope (Thermo Fisher Scientific Inc., Waltham, MA, USA). Samples were sputtered with a thin Au/Pd layer before analysis, by means of a Emitech K575X sputtering device (Quorum Technologies Ltd., Laughton, UK).

Differential Scanning Calorimetry (DSC) analyses were carried out by means of a Mettler Toledo 822 DSC (Mettler-Toledo, LLC, Columbus, OH, USA). Samples were sealed in aluminum pans and analyzed under a nitrogen flux, using the following the temperature program: heating from 25 °C to 180 °C, cooling from 180 °C to 0 °C, and heating from 0 °C to 180 °C at a heating/cooling rate of 10 °C/min. The percent crystallinity content (X_c_) was calculated according to the following equation:X_c_ = ∆Hm/∆Hm° × 100(1)
where ∆Hm is the heat of melting recorded on the sample, normalized on the content of polyethylene, and ∆Hm° is the melting enthalpy of fully crystalline polyethylene, equal to 293 J/g [11,12].

Due to the hygroscopic nature of cellulose, the physical and mechanical properties of composites containing cellulosic fillers can be influenced by the humidity absorbed onto cellulose in ambient conditions [13]. Therefore, all samples were conditioned at 25 °C and 50% relative humidity (RH) for at least 24 h before mechanical testing.

Tensile tests were carried out on dumb-bell specimens with a cross section of 4 mm^2^ using a gauge length of 26 mm and a deformation speed of 5 mm/min, by means of an Instron 4505 testing machine (ITW Inc., Glenview, IL, USA). Young’s modulus (E), peak stress (σ_max_), and elongation at break (ε_R_) were calculated from stress/strain curves as average values over at least 10 tested specimens.

Charpy impact tests were carried out on notched specimens (notch depth to width ratio of 0.3, span length 48 mm) by means of a CEAST Resil Impactor pendulum (ITW Inc., Glenview, IL, USA), equipped with a DAS 4000 Acquisition System, using an impact energy of 3.6 J and an impact speed of 0.99 m/s. Impact toughness values were calculated as average over a least five tested specimens.

Water absorption tests were carried out by immersion of samples (30 mm × 10 mm × 3 mm) in distilled water, recording the mass difference at regular intervals. All specimens were padded with dry filter paper before weighing to remove excess water from the external surfaces.

## 3. Results

### 3.1. Filler Size Distribution, Composites Preparation and Morphology

In this paper a mechanical recycling strategy of MCs has been proposed, developing materials containing up to 90 wt% of MCs in combination with recycled polyethylene. Exploring such a wide range of compositions, fully recycled materials with properties ranging from lightly-filled polymer composites to lignocellulose-based products (e.g., fiberboards) have been realized.

The size distribution of C and C-Al materials after milling was measured by sieving the powders through multiple metal sieves and recording the weight of the different fractions. The results obtained are reported in Appendix A and reveal a qualitatively similar distribution for both samples, with the maximum amount of particles retained by the 500 µm sieve. C cartons showed a larger fraction of particles with size < 200 µm (44 wt%), with respect to C-Al (31 wt%).

It is worth noting that in these systems, the size distribution of the cellulosic component is expected to change significantly during the melt processing step. In fact, the shear forces exerted by the molten polymer will induce a progressive fragmentation of paper particles, ideally approaching a separation into single cellulose fibers. A precise determination of the final particle size distribution is beyond the scope of this work, as it would require the separation of fibers from the polymeric fraction after processing. Optical micrographs of thin (100 µm) films were recorded for materials at moderate MC content, as reported in Appendix A, revealing the coexistence of both single cellulosic fibers and residual compact, paper-like cellulose clusters and, where present, Al-foil fragments.

MC was successfully mixed with PE using a conventional polymer processing apparatus and then easily molded by compression molding to obtain composites sheets. Given the polar nature of cellulosic fibers and aluminum fragments contained in MC, a very low adhesion with the PE matrix is expected. Therefore, maleated polyethylene (MAPE) was selected and added as a coupling agent. The modification of polymers with maleic anhydride polar groups is well known as an effective strategy to improve dispersion and interfacial adhesion in composites containing fillers with polar surfaces (i.e., cellulose [14,15], inorganics such as glass fibers [16], and mineral micro/nano particles [17]). Maleated polyolefins are available commercially and can be conveniently used, in appropriate contexts, as processing additives in polyolefin and recycled polyolefin-based composites [18]. The amount of MAPE was varied among 2.5 and 10 wt% (calculated with respect to the PE matrix weight) to study and optimize its effect on composite properties. At a high MC content (80 and 90 wt%), due to the high content of polar fillers and, as a consequence, high polymer/filler contact surface, MAPE was only added at the maximum percentage. Materials without MAPE were also prepared for comparison. Table 1 resumes codes and composition of all of the materials realized.

The effectiveness of the mixing procedure and the effect of the coupling agent were investigated analyzing the morphology of impact fracture surfaces by means of SEM analyses (details on impact testing are reported in Section 3.3). Micrographs of samples at the lowest and highest MC content-with and without MAPE-are reported in Figure 2 as representative examples of the composites realized.

Micrographs of samples without MAPE, at the lowest and highest MC content, show a clear separation between the polymeric fraction, deformed during the impact test, and the cellulose fibers that appear almost completely “clean” (that is, not covered by the polymer). During deformation, the matrix/fiber interfaces were largely broken, resulting in extensive debonding. In both M0 C20 and M0 C90 samples the fracture surfaces appear irregular; moreover, large fiber bundles are observed in the C90 sample, along with some single fiber, indicating a partly ineffective mixing at high MC content. In the samples containing MAPE, a different scenario is observed. The fracture surface of the M10 C20 shows fewer fibers exposed with respect to M0 C20, showing that fracture does not propagate preferentially through the matrix/fiber interfaces. This finding can be attributed to a stronger polymer/filler adhesion induced by the coupling agent. In fact, visible fibers appear well bonded and covered by a polymer layer. These considerations partly hold also for the M10 C90 sample; a polymeric layer is observed onto fibers. However, a large number of exposed fibers are also observed, indicating an improved but insufficient adhesion. In the composites based on Al-containing cartons, a similar morphology was found (see Appendix A). Morphological analyses confirm that at high MC content, the fibrous nature of the filler governs the structure of the materials.

### 3.2. Thermal Analysis

Differential scanning calorimetry was carried out on the composites to analyze the effect of fillers (cellulose, aluminum) and MAPE on the thermal behavior of the polymeric phase. The main thermal parameters obtained are reported in Table 2, while the thermograms of representative samples are illustrated in Figure 3 and in Appendix A.

The presence of MC, with or without Al, does not show a strong influence on the melting/crystallization behavior of the recycled PE matrix up to 60 wt% of the load, as inferred by the minor changes observed in the melting and crystallization temperatures. Interestingly, at 80 and 90 wt% MC, the appearance of a low temperature crystallization/melting peak was observed (Figure 3). This signal can be attributed to the phase transition of LDPE, contained in MCs (at high carton content, LDPE represents more than 50% of the polymeric phase). The presence of separated phase transitions indicates a probable phase separation of the different polyethylene species, due to their limited compatibility [19,20]. The low-temperature peak is slightly visible at 60 wt% MC (see high magnification inserts in Figure 3), but is much more evident in the 80 and 90% materials. This phenomenon is expected to reduce the homogeneity of the polymeric phase in samples at high MC load, with consequences on the mechanical behavior (as discussed in next section). The crystallinity index decreases in all samples as a function of the MC content. This is due to the geometrical constraint of the fibrous fraction that hinders polymer crystallization.

### 3.3. Mechanical Analysis

Mechanical parameters of the prepared composites were analyzed by means of tensile and Charpy impact tests. The results of mechanical testing are reported in Figure 4.

From the mechanical results, different observations can be pointed out. The recycled PE matrix has relatively high stiffness and low elongation at break, corresponding to rigid, high crystallinity HDPE grades. By increasing the MC content, the elongation (Figure 4e,f) further decreases coupled with a strong increase in elastic modulus (Figure 4a,b), up to 220% with respect to neat PE. These findings are expected as both cellulose and aluminum act as rigid fillers increasing the elastic modulus of the composites and, as a consequence, reducing ultimate elongation. The addition of MAPE slightly decreases stiffness and increases ultimate elongation in most materials. This is due to the nature of the additive backbone (LLDPE has generally lower stiffness than HDPE). At high MC content, the modulus stopped increasing in composites containing aluminum: this can be a symptom of ineffective mixing, due to the presence of large (hundreds of µm) Al-foil particles not fragmented during melt processing, as shown in Appendix A.

The peak stress of composites (Figure 4c,d) is strongly influenced by MAPE, as this parameter is more sensitive than modulus to adhesion at the polymer/filler interface [6,21]. MAPE generally led to an increase in peak stress for composites up to 60 wt% MC, in particular this effect was more significant in C-Al containing composites, that showed the maximum load-bearing ability. In contrast to what observed in previous investigations with virgin HDPE [6], in uncompatibilized materials peak stress did not decrease monotonically with increasing MC content. A possible reason is that the recycled PE matrix used in this work has developed, during service life and reprocessing, a degree of oxidation that could have slightly increased its compatibility with polar fillers. The presence of oxidation was confirmed by FTIR analysis that revealed the presence of carbonyl groups (Appendix A). The overall mechanical response of composites containing up to 60 wt% of MC falls in the range of common polyolefin composites containing lignocellulosic fillers [22,23]. For these materials, then, possible applications similar to wood-polymer and cellulose-polymer composites, which are increasingly used in the transport and construction sectors, can be foreseen [7,24].

As in the case of elastic modulus, materials at high (80, 90 wt%) MC content show a different mechanical response if compared to the low (20–60 wt%) MC content materials, with a sharp decrease of strength and a negligible effect of MAPE. At high MC load the fibrous nature of the filler dominates the structure of the material and samples at 80 and 90 wt% MC can be more effectively compared to fibrous systems with polymeric binders (such as particleboards and fiberboards) or even to cellulosic sheets (heavy paperboard) than to polymer-matrix composites. In fact, mechanical properties obtained on highly filled materials were found to be similar to those recorded on fibrous and wood-based board [25,26], and on some type of paperboard [27]. The limited effect of MAPE on the mechanical response of highly filled materials can be a consequence of the highly fibrous morphology and to the insufficient adhesion evidenced by SEM analyses.

Impact tests were carried out on notched specimens by means of an instrumented Charpy pendulum, with the recorded impact resilience reported in Figure 4g,h. Interestingly, the impact performances of pristine PE were not much affected or, in some case, slightly increased by the addition of MC (at least for contents up to 60 wt%). The recycled PE used in this work has high stiffness and low deformability, thus resulting in a low impact resilience. In these conditions, the presence of a fibrous filler effectively deviates the propagating fracture front, thus increasing the energy required to break the sample (even in the case of a weak adhesion to the polymer). The presence of MAPE generally increased the impact resilience up to 60 wt% M by increasing the energy required to separate the polar filler from the polymeric phase during fracture [28]. Materials with high MC content showed a different behavior, with a decrease of impact resilience only marginally mitigated by the coupling agent as a consequence of the fibrous morphology of these samples.

### 3.4. Water Absorption

The water absorption behavior of the realized materials was reported in terms of water uptake as function of MC content during water immersion up to 1000 h (Figure 5). Curves of weight uptake vs. immersion time of each sample are reported in the Appendix A. The water absorption recorded increases with increasing MC content, as expected due to the highly hydrophilic nature of cellulose. Water absorption was lower for materials containing C-Al at all compositions, with an uptake value that is about 14% for CAl90 compared to the 25% of C90. This large difference cannot be justified only by the lower water uptake capacity of aluminum foil compared to cellulose, but is probably also related to a barrier effect offered by the aluminum fragments at the surface of tested samples (see Appendix A). Absorption values of samples up to 60 wt% of C/C-Al are slightly lower in the presence of MAPE: this finding was attributed to the formation of a polymeric layer onto fiber surfaces, promoted by MAPE (as evidenced in Section 3.1), thus reducing their water binding tendency. At high MC content, the effect of MAPE is negligible due to high cellulose content. It is worth noting that, even at the highest MC content, water absorption kinetics and the equilibrium uptake values are generally lower than observed in typical fiberboards [29,30], especially for the composites containing C-Al, thus making these materials interesting candidates for the substitution of fiberboards in wet environments.

## 4. Conclusions

In this paper, a valorization strategy for the mechanical recycling of multilayer cartons (MCs) is reported. Industrial scraps of MCs with and without aluminum were employed in combination with recycled polyethylene to realize sustainable composites in a wide compositional range. In particular, MCs were tested as cellulosic or cellulosic/aluminum-based filler leading to the realization of thermoplastic composite materials. Good processability and formability by compression molding was demonstrated in the whole compositional range explored. Morphological analyses revealed a partial destructuration of the cellulosic component during processing, with the coexistence of both single cellulosic fibers and residual compact, paper-like cellulose clusters, and Al-foil fragments. The mechanical properties of the realized composites range from the typical values of lightly-filled polymer/wood or polymer/cellulose composites to highly fibrous materials (i.e., particleboards, fiberboards) at the highest MC content (80–90 wt%). Water absorption tests revealed a very good water resistance, in particular for the C-Al systems. Maleic anhydride modified polyethylene (MAPE) added as coupling agent during processing, was effective to improve the mechanical properties of the composites containing up to 60 wt% of MC. At higher filler content, the effect of MAPE was negligible due to the highly fibrous nature of the materials realized.

## Figures and Tables

**Figure 1 polymers-14-00807-f001:**
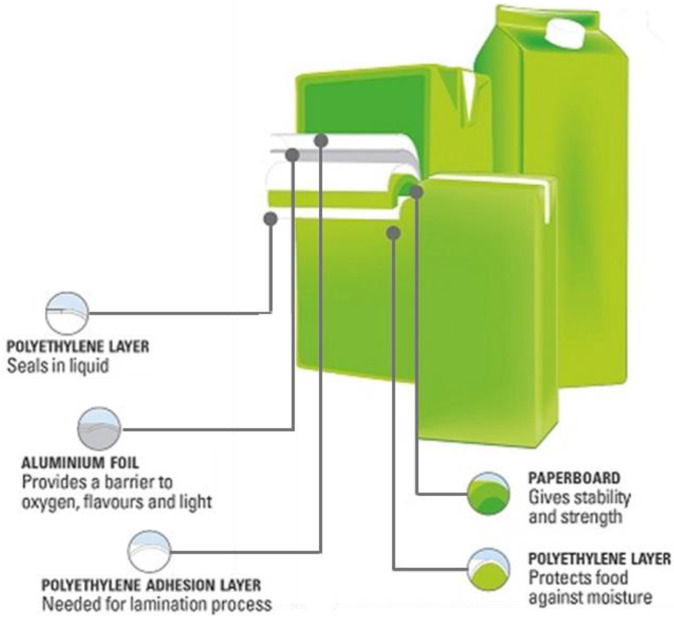
Structure of multilayer cartons, adapted from [1].

**Figure 2 polymers-14-00807-f002:**
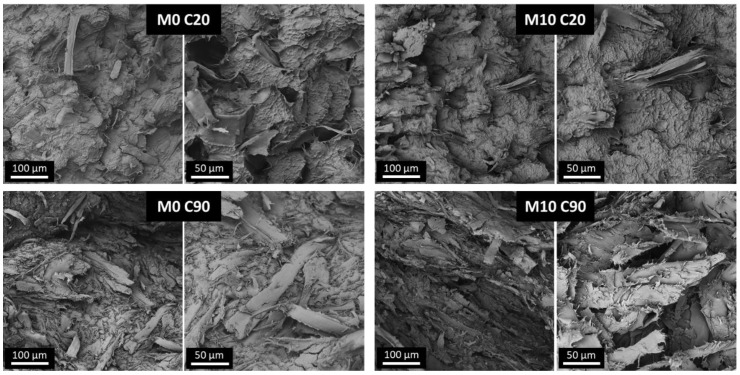
SEM micrographs of impact fracture surfaces of samples without (M0 C20, M0 C90) and with (M10 C20, M10 C90) coupling agent, at different magnification levels.

**Figure 3 polymers-14-00807-f003:**
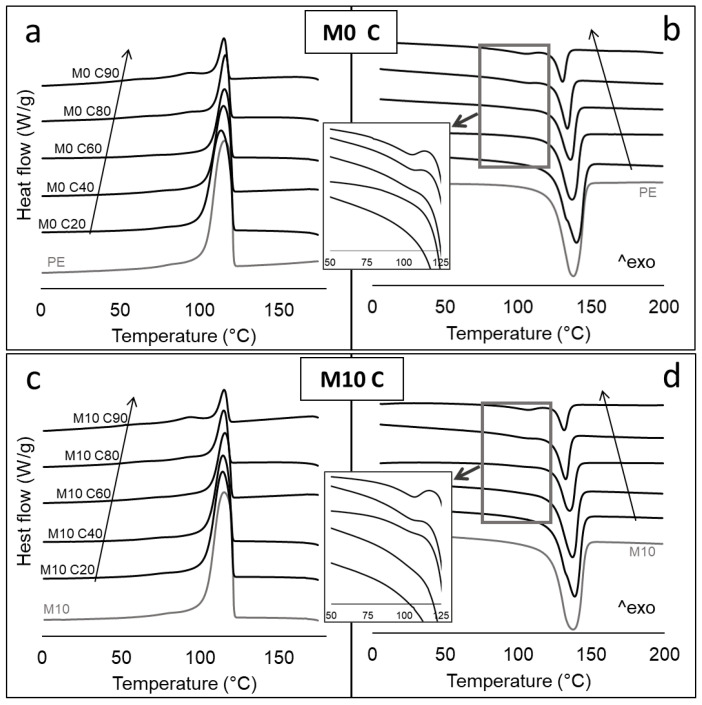
DSC thermograms showing the crystallization and melting peaks of M0 C (**a**,**b**) and M10 C (**c**,**d**) systems, respectively. Arrows indicate increasing MC content.

**Figure 4 polymers-14-00807-f004:**
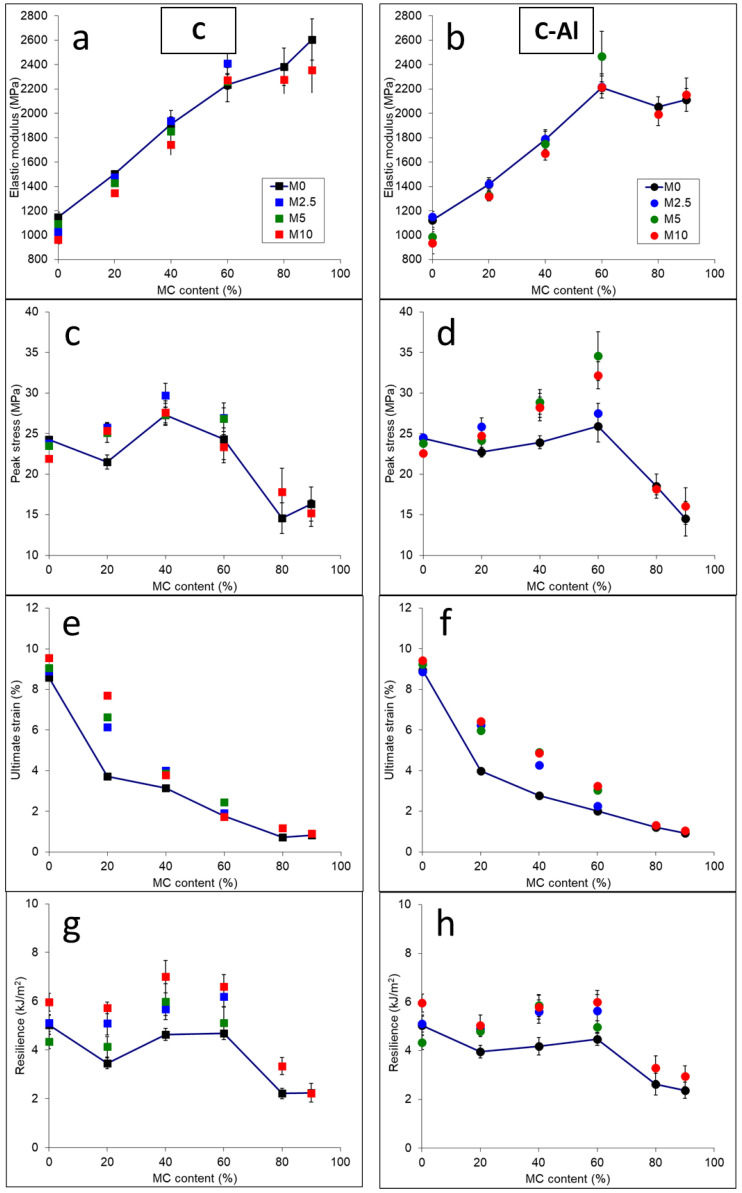
Tensile and impact parameters of the prepared composites as a function of composition: elastic modulus (**a**,**b**), peak stress (**c**,**d**), ultimate strain (**e**,**f**) and impact resilience (**g**,**h**). Lines connecting points of the M0Cx-CAlx systems are reported as a guide for the eye.

**Figure 5 polymers-14-00807-f005:**
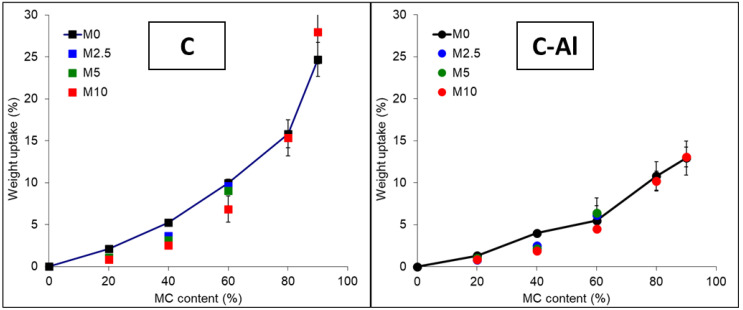
Water absorption after 1000 h of water immersion for the prepared composites. Lines connecting points of the M0Cx-CAlx systems are reported as a guide for the eye.

**Table 1 polymers-14-00807-t001:** Compositions and codes of all materials realized. Sample codes are in the format MxCy and MxCAly for cartons without and with Al-foil, respectively. The number x indicates MAPE content while y indicates carton content.

PE + MAPE	MAPE (% vs. PE)	MC (%)	Sample Codes	
			Cartons without Al (C)	Cartons without Al (C-Al)
100	-	-	PE	-
80	-	20	M0 C20	M0 CAl20
60	-	40	M0 C40	M0 CAl40
40	-	60	M0 C60	M0 CAl60
20	-	80	M0 C80	M0 CAl80
10	-	90	M0 C90	M0 CAl90
100	2.5	-	M2.5	-
80	2.5	20	M2.5 C20	M2.5 CAl20
60	2.5	40	M2.5 C40	M2.5 CAl40
40	2.5	60	M2.5 C60	M2.5 CAl60
100	5	-	M5	-
80	5	20	M5 C20	M5 CAl20
60	5	40	M5 C40	M5 CAl40
40	5	60	M5 C60	M5 CAl60
100	10	-	M10	-
80	10	20	M10 C20	M10 CAl20
60	10	40	M10 C40	M10 CAl40
40	10	60	M10 C60	M10 CAl60
20	10	80	M10 C80	M10 CAl80
10	10	90	M10 C90	M10 CAl90

**Table 2 polymers-14-00807-t002:** Results of DSC analysis of all materials realized: crystallization temperature (T_c_), melting temperature (T_m_), and crystallinity (X_c_). The crystallinity content is calculated on the basis of the polymeric content (recycled PE + the PE fraction of MC + MAPE).

Code	T_c_ (°C)	T_m_ (°C)	X_c_ (%)	Code	T_c_ (°C)	T_m_ (°C)	X_c_ (%)
PE	116	138	71	-			
M0 C20	113	140	67	M0 CAl20	113	140	66
M0 C40	115	137	59	M0 CAl40	115	137	59
M0 C60	116	136	57	M0 CAl60	116	136	56
M0 C80	116	108–133	44	M0 CAl80	115	108–132	54
M0 C90	115	106–130	36	M0 CAl90	114	106–128	48
M2.5	116	137	71	-			
M2.5 C20	113	140	66	M2.5 CAl20	116	137	71
M2.5 C40	116	136	65	M2.5 CAl40	113	140	62
M2.5 C60	115	136	54	M2.5 CAl60	116	135	59
M5	116	137	68	-			
M5 C20	114	140	67	M5 CAl20	115	138	69
M5 C40	115	137	60	M5 CAl40	114	138	54
M5 C60	116	135	56	M5 CAl60	115	136	63
M10	115	137	71	-			
M10 C20	114	139	67	M10 CAl20	115	138	64
M10 C40	114	137	62	M10 CAl40	114	138	63
M10 C60	116	135	53	M10 CAl60	115	136	59
M10 C80	115	132	42	M10 CAl80	116	109–132	54
M10 C90	115	131	31	M10 CAl90	116	106–128	45

## Data Availability

Not applicable.

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
