# Peer review of "Sustainable Cellulose-Aluminum-Plastic Composites from Beverage Cartons Scraps and Recycled Polyethylene"

_polymers, 2022, doi:10.3390/polym14040807_

Round 1

Reviewer 1 Report

The present study reports mechanical recycling approach for multilayer cartons combined with recycled polyethylene and a maleated polyethylene as coupling agent. The authors analyzed in details effect of composition on the morphology, thermal, mechanical and water absorption properties of the investigated materials. Results are presented clearly and well discussed along the document. Finally, the findings support the conclusions.  Well done ! Therefore, I recommend the acceptance of this manuscript in current form.

Author Response

We thank the reviewer for his consideration and for his encouraging words. We tried our best to make the paper comprehensive and clear, with the aim of providing a valulable reference for both the polymer and the paper recycling communities.

Reviewer 2 Report

The Comments

  • Short Abstract, needs more details.

  • P2 Line 70: what is “Al” used for? I know it is stand for “aluminum”, but it should be mentioned

  • P3 Line 99: “dividing the sample melting enthalpy by the melting enthalpy of fully crystalline polyethylene, equal to 293 J/g [11,12].” Needs explanation

  • P4 Line 140 Table 1:

Is it one sample or two sample, it is not clear

  • P9 Line 240: “Water adsorption” should be “Water absorption”

P10 Line 256:” are slightly influenced by MAPE thanks to its ability” ??

Author Response

We thank the reviewer for his careful analysis of the manuscript and for the suggestions provided. We modified and corrected the paper as requested, we are confident that the changes implemented allowed us to significantly improve the quality of the paper.

In the following lines, we report our point-by-point answer to reviewer’s comments and a description of changes made to the pristine manuscript. All changes have been made using the “Track changes” function of MS word, to assure traceability. 

Short Abstract, needs more details.

The abstract has been reworded to provide more details on the paper content (Lines 50-58 of the revised, manuscript)

P2 Line 70: what is “Al” used for? I know it is stand for “aluminum”, but it should be mentioned

The abbreviation “Al” has been defined when cited for the first time (Line 85 of the revised manuscript)

P3 Line 99: “dividing the sample melting enthalpy by the melting enthalpy of fully crystalline polyethylene, equal to 293 J/g [11,12].” Needs explanation

The calculation procedure of percent crystallinity content has been clarified and the relative equation has been reported (Lines 131-137 of the revised manuscript)

 P4 Line 140 Table 1:  Is it one sample or two sample, it is not clear

Column headers of Table 1 have been revised to clarify sample codes and compositions. Sample codes are in the format MxCy and MxCAly for cartons without and with Al-foil, respectively. The number x indicates MAPE content while y indicates carton content.

P9 Line 240: “Water adsorption” should be “Water absorption”

The error has been amended.

P10 Line 256:” are slightly influenced by MAPE thanks to its ability” ??

The entire statement has been reworded, to improve clarity (Lines 343-345 of the revised manuscript)

Reviewer 3 Report

Publication "Sustainable cellulose-aluminum-plastic composites from beverage cartons scraps and recycled polyethylene" discusses potential recycling of multilayer cartons (MC) into composite panels. Recycling of MC has challenges due to nature of layered structure, this research discusses a possibility where no layer separation is needed. Yet, there are holes in this paper that need to be addressed, mostly related to data presenting. Therefore, paper must be corrected before publication.

Author Response

We thank the reviewer for his careful analysis of the manuscript and for the suggestions provided. We modified and clarified the paper accordingly, we are confident that the changes implemented allowed us to significantly improve the quality of the paper.

In the following lines, we report our point-by-point answer to reviewer’s comments and a description of changes made to the pristine manuscript. All changes have been made using the “Track changes” function of MS word, to assure traceability.

  • Misspelling should be checked throughout the paper (like line 64 “wher” should be “where”).

A careful check of the manuscript for spelling errors and typos was carried out.

  • Overall writing/sentence structure should be checked (like line 56 – 61 is one sentence), complicated sentence structures make it difficult to read and understand the paper.

A revision of over-complicated and long sentences has been performed to improve clarity of the manuscript.

  • Composite preparation:

o obtained powders should be described (at least particle size distribution) since size will affect composite end properties significantly

We thank the reviewer for the valuable suggestion. The size distribution of milled MC samples, both with and without the aluminum layer, has been determined by sieving the materials through a stack of metallic sieves, with mesh size ranging from 1000 to 50 µm. The procedure has been inserted in the Experimental section, and results have been reported in the manuscript, in section 3.1 (lines 168-181), and in the Supplementary Material (Figures S1 and S2).

We want to underline that, though particle size distribution is an important parameter for the mechanical properties of composites, in our system the initial distribution of the cellulosic component is expected to change significantly during the melt processing step. In this phase, in fact, the shear forces exerted by the molten polymer will induce a progressive fragmentation of paper particles, ideally approaching a separation into single cellulose fibers.

A precise determination of the final particle size distribution is beyond the scope of this work, as it would require the separation of fibers from the polymeric fraction for each composition, followed by electron microscopy and size analysis. To have an idea of the presence of single cellulosic fibers and fibrous aggregates in the composites, an alternative analysis was carried out and the results have been reported in the Supplementary Material. Samples of the composites containing 40 wt% of either C and C-Al cartons were compression molded at 180°C/100 bar to obtain films (thickness of about 100 µm) and analyzed through optical microscopy. This composition was selected as the content of MC is low enough to guarantee a sufficient transparency of the materials.

Optical microscopy gave evidence of the presence of some large (> 500 µm) aggregates but also of single cellulosic fibers, though the low contrast does not allow a precise determination of fiber length and thickness. Aluminum foil, where present, is clearly observed and fragmented in irregular flakes.

  • pelletization process needs to be described more specifically (pressure, duration)

A detailed description of the pelletization and molding procedure was added to the manuscript (Lines 113-118 of the revised manuscript)

  • All of the used equipment should be described as Phenom compact scanning electron microscope (with full manufacturer information given in line 94 brackets).

Complete producer details have been added for all instruments used.

  • Same magnification images of SEM should be provided for more clear comparison of the data (Figure 2).

SEM micrographs at the same magnification level have been reported for all samples.

  • When multiple graphs/figures used in one Figure abbreviations (a), (b), etc. should be used (easier to refer in text).

Reference letters have been added on figures of DSC thermograms and mechanical parameters. For SEM micrographs, the presence of labels and size marks was deemed enough to clarify the content of the different panels of the figures.

  • Abstract mentions that obtained recycled composites show properties ranging from lightly-filled polymer composites to rigid, fiberboard-like panel. Obtained data should be compared with data describing these lightly-filled polymer composites and rigid, fiberboard-like panels. As well as potential application of these obtained recycled panels should be mentioned.

The reference to lightly filled polymer composites, holding for compositions with MC content up to 60 wt%, is a generic description of the materials realized. There are countless examples of composites containing cellulosic or ligno-cellulosic fillers, with a wide range of properties, so that a detailed comparison of properties was not considered meaningful. Some example of composites has been cited as a reference (n. 22, 23, 24 of the revised manuscript). Possible applications fields have also been mentioned, based on literature (Lines 295-299 of the revised manuscript).

 As for the composites with higher MC content, the comparison with fiberboards is a natural one considering the high amount of cellulose with respect to the polymeric phase. For these materials references were cited (26, 27) containing relevant examples of the mechanical and water absorption properties showed by particleboard and fiberboard materials.

  • Conclusions need to be improved: more what was learned in this research and what does it tell us.

Conclusions have been revised including a more detailed discussion of the main results and findings.

Reviewer 4 Report

Comment Number 1: Corresponding to unyielding, high crystallinity HDPE grades, at break, the castoff PE matrix has relatively high stiffness and low elongation by increasing the MC content, the extension further diminishes paired with a robust boost in elastic modulus. These discoveries are anticipated as both cellulose and aluminum act as rigid grouts rising the elastic modulus of the composites. Comment Number 2: At tall, MC load the gristly nature of the caulking governs the structure of the material and illustrations at 80 and 90 wt% MC can be more effectually compared to fibrous systems with polymeric binders or even to cellulosic sheets than to polymer-matrix composites. Comment Number 3: Thermoplastic polymers and polyolefins in precise are the most explored matrices for such composites, nonetheless, various polymers still can be suggested, including thermosets. The ability of the selected polymer to form strong, due to the highly polar and hydrophilic nature of cellulosic materials, well adhered interfaces with such fillers is a crucial element to define processing and additivities strategies in further research. It’s a good research direction for future research.

Author Response

We thank the reviewer for his analysis and for the useful comments provided. We inserted its observation in the paper, improving clarity of our discussion.

Comment Number 1: Corresponding to unyielding, high crystallinity HDPE grades, at break, the castoff PE matrix has relatively high stiffness and low elongation by increasing the MC content, the extension further diminishes paired with a robust boost in elastic modulus. These discoveries are anticipated as both cellulose and aluminum act as rigid grouts rising the elastic modulus of the composites.

A comment was added to the discussion of mechanical tests (line 285 of the revised manuscript)

Comment Number 2: At tall, MC load the gristly nature of the caulking governs the structure of the material and illustrations at 80 and 90 wt% MC can be more effectually compared to fibrous systems with polymeric binders or even to cellulosic sheets than to polymer-matrix composites.

A comment was added to the morphological analysis section (lines 135-137 of the revised manuscript)

Comment Number 3: Thermoplastic polymers and polyolefins in precise are the most explored matrices for such composites, nonetheless, various polymers still can be suggested, including thermosets. The ability of the selected polymer to form strong, due to the highly polar and hydrophilic nature of cellulosic materials, well adhered interfaces with such fillers is a crucial element to define processing and additivities strategies in further research. It’s a good research direction for future research.

We agree that using different polymers can lead to interesting results, a brief comment on that was given in the introduction. This approach will be the basis for future developments on the treatment of polymer/paper packages.

Round 2

Reviewer 3 Report

The manuscribt "Sustainable cellulose-aluminum-plastic composites from beverage cartons scraps and recycled polyethylene" discusses potential recycling of multilayer cartons (MC) into composite panels. Recycling of MC has challenges due to nature of layered structure, this research discusses a possibility where no layer separation is needed. The authors have carried out enough characterization and analysis with a good discussion on the results obtained. The authors have responded to the previous reviewers fairly and made necessary changes to the manuscript. The oweral technical aspects  of the manuscribt have been improved. The reviewer believe that no further improvements are needed before this manuscribt can be published.